# Effects of Free-Range Systems on Muscle Fiber Characteristics and Welfare Indicators in Geese

**DOI:** 10.3390/ani15030304

**Published:** 2025-01-22

**Authors:** Guoyao Wang, Jianzhou Chen, Yujiao Guo, Kaiqi Weng, Yu Zhang, Yang Zhang, Guohong Chen, Qi Xu, Yang Chen

**Affiliations:** 1College of Animal Science and Technology, Yangzhou University, Yangzhou 225009, China; wgy15050786372@163.com (G.W.);; 2Joint International Research Laboratory of Agriculture and Agri-Product Safety, The Ministry of Education of China, Yangzhou University, Yangzhou 225009, China; 3Key Laboratory for Evaluation and Utilization of Poultry Genetic Resources of Ministry of Agriculture and Rural Affairs, Yangzhou University, Yangzhou 225009, China

**Keywords:** free-range systems, muscle fiber characteristics, animal welfare, goose, behaviors

## Abstract

Free-range systems are among the most important factors affecting animal health and welfare, with free-range systems offering more opportunities for animals to perform a wide range of behaviors. This study shows that free-range systems not only improve muscle fiber characteristics in geese but also significantly improve their welfare, including feather cleanliness, gait, and natural behavior. In addition, geese raised in free-range environments exhibited healthier growth and development than those raised in closed systems. These findings provide important insights for optimizing free-range farming methods and establishing a foundation for future research on enhancing poultry farming through improved animal welfare and meat quality.

## 1. Introduction

Highly industrialized animal production systems have severely impacted the natural behavior and welfare of animals, resulting in various abnormal behaviors and physical injuries [1]. Housing systems significantly affect the natural behavior, product quality, health, and welfare of animals [2,3,4]. Amidst the growing public awareness of animal welfare, the consumption of poultry meat from free-range systems has increased [5].

A free-range system is a method for raising poultry in an open-air environment, allowing for additional space for exercise and natural behavior [6]. Under this system, domesticated animals can exercise more and gain access to fresh and different types of plant-based foods [7]. The free-range system promotes improved meat quality; lambs subjected to pasture systems presented slaughter weights similar to those of animals kept in confinement, but they had high-quality meat with mild tastes and odors [8]. Chickens with free-range access were lighter at 70 days of age than those raised indoors, and they exhibited a higher percentage of polyunsaturated fatty acids than those fed indoor meat [9]. In geese, free-range systems have been shown to enhance thigh muscle weight, lower pH, and reduce moisture content [10].

Importantly, the free-range system positively contributes to animal welfare. The free-range system provided hens with more opportunities to engage in movement, comfortable behavior, and social interactions [7]. Furthermore, hens in the free-range system exhibited better plumage conditions and lower rates of footpad dermatitis [11]. In dairy cows, pastures allow grazing and can facilitate the expression of lying, standing, walking, and estrous behaviors [12]. In addition, pastures can decrease negative social interactions between cows, possibly because more space is provided per cow than is normally available indoors [13].

Although the benefits of free-range systems for animal welfare are well-established, it is unclear whether this is true of geese. Moreover, the free-range system allows ample space for exercise; however, whether it affects muscle fiber properties in animals is poorly understood. Therefore, we hypothesized that free-range systems might also contribute to improving animal welfare and muscle fiber traits. In this study, we investigated the differences in the muscle fiber characteristics and welfare indicators of Yangzhou geese raised under two different pasture systems and an indoor system, which may help goose farmers and animal scientists make informed decisions regarding their housing systems and welfare.

## 2. Materials and Methods

### 2.1. Animals, Experimental Design, and Diets

All animal experiments were approved by the Animal Care and Use Committee of Yangzhou University. Three housing systems were used: long-distance pasture system (LDPS; 200 m), short-distance pasture system (SDPS; 50 m), and indoor system (IS). This experiment was conducted at the Yang Guangyulu Family Farm, located in Dantu District, Zhenjiang City, China. A total of 270 healthy, 28-day-old male Yangzhou geese with similar weight (1.61 ± 0.13 kg) were randomly selected from the same batch and divided into the three systems (Figure 1). Under the SDPS, geese were allowed to move freely in an area 50 m away from the shed from 9:00 am to 4:00 pm, had free entry and exit to the shed, and stayed in the shed when resting. Under the LDPS, the geese were allowed to move freely in an area 200 m away from the shed from 9:00 am to 4:00 pm, also with free entry and exit to the shed, and remained in the shed when resting. Under IS, geese were raised in a shed containing feeders and waterers.

The three systems each feature identical indoor sheds, measuring 3.2 × 6.3 m, and are equipped with feeders and waterers. The geese were raised to 70 days of age. Geese in the pasture systems grazed freely from 9:00 to 4:00 every day. The number of steps taken daily was recorded using a pedometer and signal receiver. To comprehensively evaluate the behavior of geese, six cameras were installed in each pen to continuously monitor their activities. These cameras were strategically positioned to capture the entire living area, ensuring that all relevant behaviors were clearly visible and accurately recorded. The sheds were cleaned daily, with sufficient ventilation and natural light provided. During the feeding period, all the geese were provided with the same diet and had ad libitum access to food and water. The ingredients and nutritional levels of the basal diets are listed in Table 1.

### 2.2. Hematoxylin–Eosin Staining (HE) and Immunohistochemistry (IHC)

Yangzhou geese were slaughtered at 70 days of age after a 12 h fasting period. The geese were captured and transported to the laboratory within 1 h. The animals were anesthetized with sodium pentobarbital and euthanized by manual bleeding. Six male geese were randomly selected from each of the three systems for the collection of breast muscle (BM) and leg muscles. Among the leg muscles, the gastrocnemius (GAS), soleus (SOL), and extensor digitorum longus (EDL) muscles (0.5 × 0.5 × 1.0 cm) were separated. The samples used for sectioning were preserved in 4% paraformaldehyde. Each muscle sample was fixed in 4% paraformaldehyde for 24 h, embedded in paraffin, and sliced into 5 mm sections. These sections were stained with HE. For IHC analysis, samples underwent xylene and ethanol treatments and were then incubated overnight at 4 °C with primary antibodies targeting fast (MYH1A, 1:1200 dilution, ab51263; Abcam, Cambridge, UK) and slow (MYH7B, 1:4000 dilution, ab11083; Abcam, Cambridge, UK) myosin skeletal chains. After washing with phosphate-buffered saline, the slides were exposed to rabbit anti-mouse IgG secondary antibody (1:200 dilution, ab6728; Abcam, Cambridge, UK) for 30 min, washed again, and stained with 3,3-diaminobenzidine for 10 s. Samples scanned with NanoZoomer (Hamamatsu, Sydney, Australia) were analyzed using Image Pro Plus 6.0 to determine fiber diameter, cross-sectional area, density, and endo/perimysium thickness. Three random fields, each with 100 fibers, were examined for each sample.

### 2.3. Feather Scoring

The feather scores included cleanliness and degree of damage. The feather cleanliness and degree of damage scores of the geese were modified according to the feather cleanliness assessment standard developed by Mahmoud et al. [14] and the feather damage assessment standard developed by Wechsler et al. [15]. At 69 days of age, the back and chest feathers of six male geese from each group were observed and scored. Feather cleanliness was scored in three parts of the body: the back, wings, and chest and abdomen. The feathers were observed for dirt, dust, feces, etc., and the score was given according to the size of the dirty area. The degree of feather damage was evaluated by evaluating the feathers on the head, neck, chest and abdomen, wings, back, legs, tail, and anus, and the exposed skin area of each part was observed and scored. The specific scoring standards are presented in Table 2. To ensure the objectivity of the scoring, each goose is scored by three different observers.

### 2.4. Gait Scoring

The goose gait score was modified based on gait stability evaluation standards developed by Jones et al. [16]. At 69 days of age, the gaits of each male goose in each group were observed and scored individually. The gait of the geese was observed from 8:30 to 11:00 a.m. and from 2:00 to 4:30 p.m. using camera recordings. To ensure objectivity, each system was assessed by three observers during the same time period. The scoring standards are listed in Table 3. Following the gait evaluation, the percentage distribution of each gait grade was calculated for each repetition.

### 2.5. Behavioral Indicators

The behavioral indicators of geese and the definitions of each behavior were developed and modified based on van Krimpen et al. [17]. These behaviors, including standing, lying down, feeding, drinking, feather combing, feather pecking, wing flapping, and single-foot standing, are presented in Table 4. Observations were conducted for 2.5 h each in the morning and afternoon, and the frequency of each behavior was recorded. At 63 days of age, six male geese were randomly selected from each group for behavioral observations for one week. Six cameras were used to record the frequency of various behaviors in each group of geese daily from 8:30 to 11:00 am and from 2:00 to 4:30 pm. To avoid the impact of humans or other external factors on geese behavior, only the middle 1.5 h of the video data was selected for observation. Additionally, three observers manually recorded the frequency of each behavior by reviewing the video clips, ensuring consistency and accuracy in data collection. The collected data were then compiled and analyzed to assess the impact of different feeding systems on the behavioral patterns of the geese.

### 2.6. Statistical Analysis

The data were initially organized in Excel to create a database, and statistical analyses were conducted using SPSS (version 17.3, IBM, Amunk, NY, USA). Before the analysis, the data were tested for normality and homogeneity of variance. The Yangzhou goose step count was subjected to a one-way ANOVA followed by the Duncan multiple range test. Linear mixed models were employed to examine the relationships between different feeding methods (LDPS, SDPS, and IS), muscle fiber characteristics, gene expression levels, and welfare indicators (feather score, gait score, and behavioral frequency). In the analysis of muscle fiber characteristics, free-range systems and muscle type (BM, GAS, SOL, and EDL) were treated as the main effects, and individual animals within each group were treated as random effects. There was a significant interaction between the pasture system and muscle tissue for the differences in fiber density; there was no interaction in the fiber cross-sectional area, fiber diameter, thickness of endomysium, and thickness of perimysium. For welfare indicators, free-range systems were used as the main effects, whereas individuals within groups were treated as random effects. Results are presented as mean ± standard deviation, and *p* < 0.05 is considered statistically significant. The significance level is set at 0.05, which corresponds to a 95% confidence interval. The F-value is used to test whether there are significant differences in the fixed effects (free-range systems), while the *p*-value is used to determine the statistical significance of these differences. Results are presented as mean ± standard deviation, and *p* < 0.05 is considered statistically significant.

## 3. Results

### 3.1. Average Daily Steps of Geese Under Different Systems

The daily step count of the geese in the pasture system was evaluated using a pedometer. As shown in Table 5, the number of steps taken by the geese in the free-range systems was significantly higher (*p* < 0.05) than that in the IS. Additionally, the average daily step count of the geese in the LDPS was 5776.97, which was significantly higher than that of the geese in the SDPS.

### 3.2. Comparison of Muscle Fiber Morphological Characteristics of Geese Under Different Systems

To evaluate the effects of free-range systems on the physical and chemical properties of muscle fibers, we first stained sections of the BM and leg muscles (including the SOL, EDL, and GAS) using HE (Figure 2). We measured muscle fiber density, cross-sectional area, diameter, and endomysium and perimysium thicknesses in these muscles. A significant interaction was observed between the pasture system and muscle tissue owing to differences in fiber density (Figure 3A). The density of BM fibers was significantly higher (*p* < 0.05) than that of other muscle tissues in all three systems (Figure 3A). The diameter of the SOL muscle in free-range geese was significantly larger (*p* < 0.05) than that in IS, whereas the diameters of the EDL and GAS muscles in LDPS were significantly larger (*p* < 0.05) than those in IS and SDPS (Figure 3C). In addition, the perimysium thickness of the EDL of geese in the SDPS group was significantly thicker (*p* < 0.05) than those of geese in the LDPS and IS groups (Figure 3E). Based on the above results, the muscle fibers of geese in the pasture system were more developed, and the diameters of the EDL and GAS of geese in LDPS were thicker than those in SDPS.

### 3.3. Comparison of Muscle Fiber Composition of Geese Under Different Systems

To investigate the myosin heavy chain (MyHC) muscle fiber type in free-range systems, we performed IHC analyses using anti-fast MyHC (MYH1A)-stained type IIB muscle fibers and anti-slow MyHC (MYH7B)-stained type I muscle fibers for breast and leg muscles (SOL, EDL, and GAS). The BM across all systems was composed only of fast-twitch muscle fibers (Figure 4). In the EDL, the proportion of fast-twitch fibers in the LDPS was significantly higher than in the IS (*p* < 0.05). In the SOL, the proportion of slow-twitch fibers in the SDPS was significantly greater than in both the LDPS and IS (*p* < 0.05). A small amount of slow-twitch fibers was also observed in the EDL and GAS. Overall, the different systems had no effect on the muscle fiber types, and the muscles of 70-day-old geese were primarily composed of fast-twitch fibers.

### 3.4. Comparison of Feather Score of Geese Under Different Systems

To further evaluate the welfare of geese under the different systems, we compared welfare indicators, including feather quality, walking ability, and behavioral changes. First, the feather cleanliness of geese in different systems was determined (Figure 5). The feathers of the chest and back of geese in the LDPS group were significantly cleaner (*p* < 0.05) than those of geese in the IS group, whereas no significant difference in cleanliness was observed for the wing feathers of geese across all systems (Table 6). Additionally, we analyzed the grayscale values of the images in ImageJ (Fiji, National Institute of Health, Bethesda, MD, USA, https://imagej.net/ij/) (Appendix A), which confirmed the consistency of these findings. Furthermore, the degree of feather damage in the LDPS (1.29 ± 0.09) and SDPS (1.97 ± 0.09) groups was lower (*p* < 0.05) than that in the IS group (3.66 ± 0.06). Interestingly, there was no significant difference in the degree of feather damage between the geese in the LDPS and SDPS groups.

### 3.5. Comparison of Gait Scores of Geese Under Different Systems

The gait scores of the geese in the different systems are listed in Table 7. Different pasture systems significantly affected the proportion of geese with scores of 2 (*p* = 0.021), 1 (*p* = 0.018), and 0 (*p* = 0.023). Among the geese in the IS group, the proportion of geese with a gait score of 0 (able to walk normally) was significantly lower (F = 18.71, *p* < 0.05) than that of geese in the LDPS and SDPS groups. The opposite was observed for the proportion of geese with gait scores of 1 (slight gait defects) and 2 (obvious gait defects). In addition, the pasture system had no significant effect on the proportion of geese with gait scores of 3 (barely walking under an external force) or 4 (unable to walk with their feet).

### 3.6. Comparison of Behavioral Changes in Geese Under Different Systems

The behavioral changes in the geese under the different systems are presented in Table 8. The systems had a significant effect on the single-foot standing and feather pecking of the geese. The pecking frequency of geese in the IS was significantly higher (F = 22.10, *p* < 0.05) than that in the pasture system, whereas the frequency of single-foot standing was significantly higher (F = 8.34, *p* < 0.05) in the pasture system than in the IS. In addition, no significant differences in the frequency of standing, lying down, eating, drinking, combing feathers, or wing flapping were observed among the different systems. In particular, there was no significant difference in behavioral frequency between geese in the LDPS and SDPS groups. These results suggest that pasture systems allow geese to perform natural behaviors, such as single-foot standing, feather pecking, and walking, regardless of grazing distance.

## 4. Discussion

Global consumer demand drives producers to prioritize free-range systems [18,19]. These systems have a positive impact on the natural behavior of geese, allowing them to move freely and forage in outdoor spaces, which significantly enhances their welfare. Additionally, these systems improve the quality and flavor of goose meat, meeting consumer demands for high-quality poultry products. Our previous research revealed that the free-range system did not significantly influence the dressing percentage, percentage of partially eviscerated weight, and percentage of formally eviscerated weight in geese. But it notably increased the proportion of leg muscle weight while significantly reducing both the pH value and moisture content [10]. However, the muscle fiber characteristics and welfare indicators of geese under different free-range systems remain unclear. This study investigated the effects of different systems (LDPS, SDPS, and IS) on the muscle fiber characteristics and welfare indicators (feather quality, walking ability, and behavioral changes) of Yangzhou geese, providing valuable insights for commercial goose production.

Free-range systems (LDPS and SDPS) offer space for geese to exercise and interact with other individuals [20]. In this study, we compared the average number of daily steps taken by geese in different systems. As expected, grazing distance significantly affected the number of steps taken. The number of steps taken by the geese in the IS system was significantly lower than that in the pasture system. Numerous studies have reported that exercise results in skeletal muscle growth [21,22]. However, further evidence is required to determine how different systems influence and shape muscle fibers. Our study revealed that the muscle fiber diameter increased as the exercise amount increased. Yin et al. [23] found that with increased exercise, the abundance of mRNA and protein of the muscle regulatory factors *MyoD1* and *Myf4* in the skeletal muscle of adult chickens increased, leading to an increase in muscle fiber diameter. Meat tenderness, an important factor for consumers, is influenced by the muscle fibers, perimysium, and endomysium thickness [24]. However, studies on the effects of exercise on the perimysium and endomysium in poultry are limited. Our study revealed that moderate exercise contributed to the thickness of perimysium, thereby enhancing meat quality. In terms of muscle fiber composition, the proportion of slow-twitch fibers in the SOL muscle of geese under SDPS was significantly higher than that under LDPS and IS. During endurance or aerobic exercise, succinate induces the transformation of skeletal muscles from fast-to-slow-twitch fibers via *SUNCR1* [25]. Previous studies have shown that exercise can promote an increase in the proportion of type I muscle fibers (slow-twitch fibers) [26]. Generally, a higher proportion of type I fibers enhances meat tenderness and color, leading to superior meat quality [27]. However, in the present study, the different systems had no impact on the muscle fiber types, and the goose muscles were primarily composed of fast-twitch fibers. Type I, or slow-twitch fibers, are seen in high abundance during the early stages of life. However, the proportion of fast-twitch fibers increases substantially with rapid development. This shift occurs because the oxidative metabolic capacity declines with age, and the body compensates for this metabolic decline by increasing the number of fast-twitch fibers [28]. The housing system and feeding environment in later life stages may not significantly change the type of muscle fibers produced. In summary, the texture and flavor of goose meat may be slightly enhanced by using a suitable free-range system.

Intensive poultry farming practices can greatly increase meat production; however, they severely restrict poultry movement and adversely affect animal welfare (comfort levels and natural behavior) [29]. In the present study, we found that geese in free-range systems performed better in terms of feather quality, walking ability, and behavioral traits.

Geese under IS were more susceptible to feather damage, highlighting the negative impact of restricting animal movement on feather quality. Additionally, we observed a decrease in feather cleanliness in the chest, abdomen, and back regions as goose activity levels decreased. Adequate exercise promotes blood circulation and metabolism, accelerates feather growth and development, and results in denser and lustrous feathers [30]. Current research on poultry farming indicates that restricting the activity space for broiler and layer chickens leads to poor feather quality [31]. In contrast, increasing activity space has been shown to improve feather quality [32]. Therefore, it is important for geese to have access to pastures to maintain feather quality.

Gait scoring remains a valuable tool and is considered the classic method for determining leg health. Moderate exercise promotes the development and strength of the thigh muscles in poultry, thereby enhancing walking and activity [33]. Studies on egg-laying and meat-producing chickens have shown a significant impact of exercise on walking ability [16,34]. In the present study, the number of geese under IS exhibiting gait issues (mild and severe) was higher than that of geese with SDPS and LDPS, whereas the proportion of geese with a normal gait showed a decreasing trend. Geese under IS have limited space for activity, which restricts natural behaviors, adversely affecting their walking ability and resulting in decreased exercise levels. Therefore, for poultry, particularly waterfowl, ensuring an adequate area to maintain their ability to walk normally is essential.

Behavior is crucial to the manner in which an animal adapts to its physiological state and environment. In the present study, the frequency of feather-pecking behavior in geese was significantly higher in the IS than that in the pasture systems. Studies have reported that increasing the activity range of poultry helps to reduce the occurrence of feather-pecking behavior [35,36]. This is consistent with our findings, indicating that when space for geese activity is restricted, they exhibit feather-pecking behavior more frequently. The frequency of one-legged standing behavior, as a measure of relaxation, was significantly higher in the pasture systems than that in the IS. Therefore, we speculate that the caged environment limited the activity space of the geese, thereby inhibiting one-legged standing behavior. Additionally, geese may require one-legged standing to relax because of the increased exercise in free-range systems. Therefore, ensuring an appropriate housing system is essential for poultry farming. This research presented in this paper currently provides a reference for improving both meat quality and animal welfare in geese breeding. However, free-range systems may be associated with higher costs, which burden producers and hinder industry growth [37]. It is urgent to explore cost-effective farming systems that maintain both the quality and health of poultry meat while reducing production expenses. Further studies are needed to determine how to achieve a win-win outcome in production. Moving forward, we will focus on the economic cost-effectiveness of free-range systems, including factors such as land costs and equipment maintenance, as well as economic traits like meat production and down yield.

## 5. Conclusions

The results of this study provide sufficient evidence that geese under a free-range system have more developed muscle fibers without altering their muscle fiber type. In addition, free-range systems contribute to improved feather cleanliness and reduced damage. The pasture system also allowed the geese to perform natural behaviors, such as single-foot standing, feather pecking, and walking, regardless of grazing distance. Therefore, a free-range pasture system is an effective measure to improve the welfare of geese.

## Figures and Tables

**Figure 1 animals-15-00304-f001:**
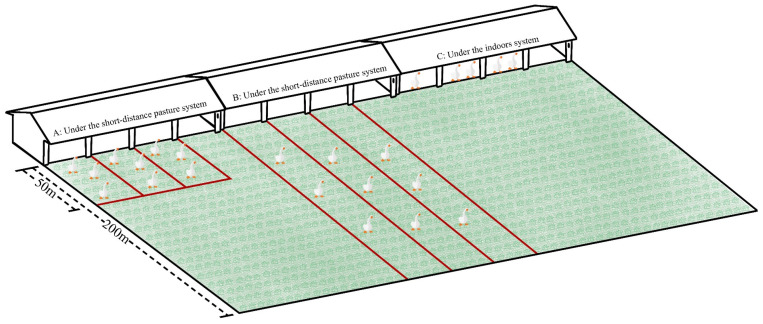
Layout of the experimental design. (**A**) The short-distance pasture system (SDPS; grazing distance of approximately 50 m) in the open area from 9:00 am to 4:00 pm; (**B**) the long-distance pasture system (LDPS; grazing distance of approximately 200 m) in the open area from 9:00 am to 4:00 pm; (**C**) indoor system (IS) in the shed. The red line indicates the fence.

**Figure 2 animals-15-00304-f002:**
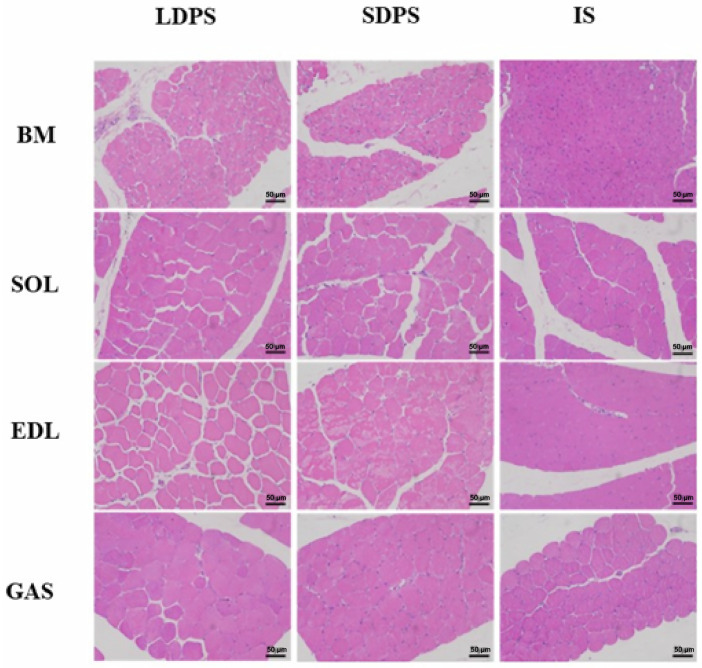
HE sections of muscle fibers of Yangzhou geese with different exercise amounts. Morphological analysis of different muscle tissues of geese in the free-range systems. BM, breast muscle; SOL, soleus; EDL, extensor digitorum longus; GAS, gastrocnemius; LDPS, long-distance pasture system; SDPS, short-distance pasture system; IS, indoor system. Scale bar = 50 µm.

**Figure 3 animals-15-00304-f003:**
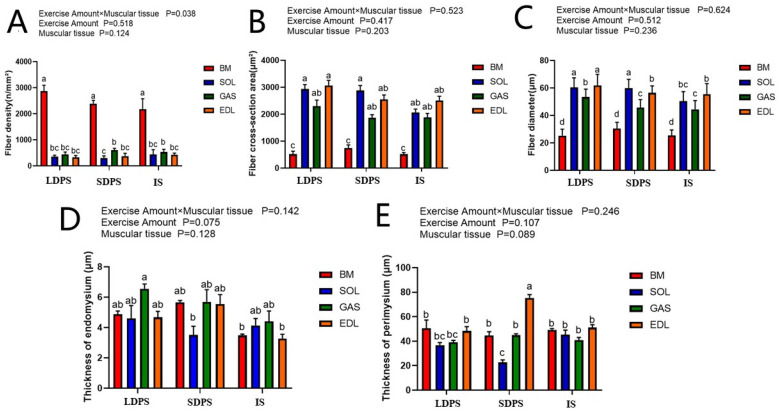
Effects of free-range systems on muscle fiber density (**A**), cross-sectional area (**B**), diameter (**C**), endomysium (**D**), and perimysium (**E**) thickness of Yangzhou geese. BM, breast muscle; SOL, soleus; EDL, extensor digitorum longus; GAS, gastrocnemius; LDPS, long-distance pasture system; SDPS, short-distance pasture system; IS, indoor system. Statistically significant differences are indicated by different letters.

**Figure 4 animals-15-00304-f004:**
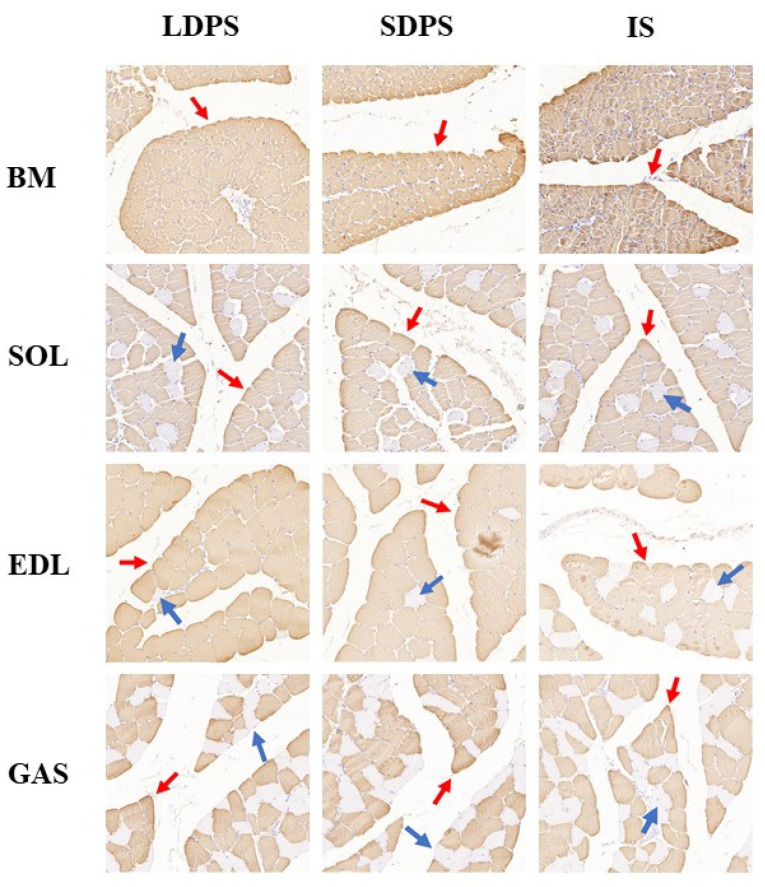
The muscle fiber type composition analysis in geese in the different systems using immunohistochemical of anti-fast MyHC (MYH1A) and anti-slow MyHC (MYH7B). Red and blue arrows point to examples of fibers with positive immunostaining for fast-twitch and slow-twitch myosin, respectively. BM, breast muscle; SOL, soleus; EDL, extensor digitorum longus; GAS, gastrocnemius; LDPS, long-distance pasture system; SDPS, short-distance pasture system; IS, indoor system.

**Figure 5 animals-15-00304-f005:**
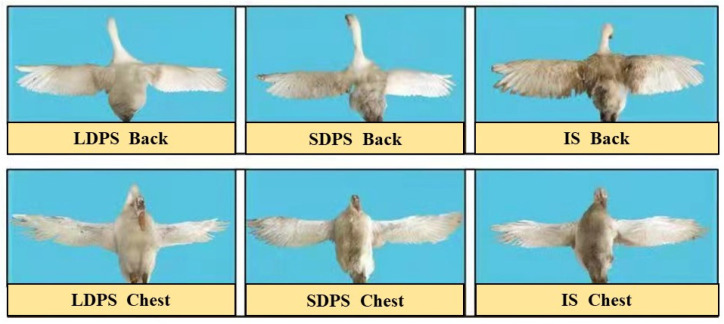
Effect of systems on the cleanliness of goose feathers. BM, breast muscle; SOL, soleus; EDL, extensor digitorum longus; GAS, gastrocnemius; LDPS, long-distance pasture system; SDPS, short-distance pasture system; IS, indoor system.

**Table 1 animals-15-00304-t001:** Ingredients and nutrient level of basal diet.

Component	Percentage (%)	Nutrient Level	Content
Maize	57.0	CP	15.0%
Soybean meal	20.0	CEE	4.31%
Wheat bran	14.0	CF	8.0%
Premix ^1^	5.5	CA	8.0%
Bone meal	3.5	AP	0.53%
		Lys	0.28%
		Met	0.23%
		ME	11.52 MJ/kg

Note: ^1^ Premix provided per kilogram of diet: vitamin A, 9000.0 lU; vitamin D3, 3000.0 lU; vitamin E, 24.0 lU; vitamin B12, 0.1 mg; vitamin K3, 1.8 mg; riboflavin, 5 mg; pantothenic acid, 15 mg; pyridoxine, 3 mg; biotin, 0.05 mg; niacin, 40.0 mg; choline, 500.0 mg; biotin, 0.05 mg; pyridoxine, 3.0 mg; riboflavin, 5.0 mg; manganese, 80.0 mg; zinc, 90.0 mg; iron, 80.0 mg; iodine, 0.35 mg; copper, 20 mg; and selenium, 0.3 mg.

**Table 2 animals-15-00304-t002:** The scoring standard for feather quality in geese.

Score	Feather Cleanliness	Feather Damage
0	Completely clean	Feathers are of good quality and have no damage
1	The dirty area is less than 1/4 of the scoring area	Feathers are damaged, but no skin is exposed
2	The dirty area accounts for 1/4–1/3 of the scoring area	The exposed skin area is less than 3 cm × 3 cm
3	The dirty area accounts for 1/3–1/2 of the scoring area	The exposed skin area is more than 3 cm × 3 cm
4	The dirty area is greater than 1/2 of the scoring area	The skin is completely exposed

**Table 3 animals-15-00304-t003:** The scoring standard for geese gait.

Score	Gait
0	The test geese can easily walk more than ten steps and maintain their balance well.
1	The experimental geese have a slight gait defect that is difficult to detect and requires careful observation to see.
2	Experimental geese had obvious gait defects—their legs were shaky, unsteady, or limp when walking.
3	The test geese can only walk reluctantly when driven by external forces and generally do not move when there is no external force.
4	The experimental goose can no longer walk on its feet and can only move by flapping its wings.

**Table 4 animals-15-00304-t004:** The definition for behavior traits in geese.

Types of Behavior	Definition
Standing	The goose rises from the floor and maintains a vertical position with its legs erect for at least 2 s
Lying down	Goose resting on the floor
Feeding	Goose’s head reaches into the trough to feed
Drinking	Goose’s head reaches into the drinking fountain to drink water
Feather combing	The goose preens its feathers with its beak for at least 5 s
Feather pecking	Peck other geese’s feathers quickly and vigorously
Wing flapping	The goose flaps its wings at least twice while in motion or at rest
Single-foot standing	Goose stands on only one foot

**Table 5 animals-15-00304-t005:** Average daily steps of geese under different systems.

Group	The Pasture System		IS
	LDPS	SDPS	
Average steps per day	5776.97 ± 560.70 ^a^	4520.17 ± 399.93 ^b^	2735.76 ± 440.42 ^c^

Note: Average daily steps of Yangzhou geese in the long-distance pasture system (LDPS), short-distance pasture system (SDPS), and indoor system (IS). In the same row, values with different superscripts lowercase letters indicate significant differences (*p* < 0.05).

**Table 6 animals-15-00304-t006:** Feather cleanliness score of geese under different systems.

Body Parts	LDPS	SDPS	IS	F Value	*p* Value
Back	1.68 ± 0.57 ^b^	2.01 ± 0.06 ^ab^	2.33 ± 0.59 ^a^	2.02	0.038
Chest	1.07 ± 0.03 ^b^	1.34 ± 0.51 ^b^	2.13 ± 0.10 ^a^	7.58	0.027
Wing	1.67 ± 0.58	1.69 ± 0.49	2.25 ± 0.38	4.15	0.332

Note: Feather cleanliness score of Yangzhou geese in the long-distance pasture system (LDPS), short-distance pasture system (SDPS), and indoor system (IS). In the same row, values with different superscripts lowercase letters indicate significant differences (*p* < 0.05).

**Table 7 animals-15-00304-t007:** The proportion of geese that obtained corresponding gait scores under different systems (%).

Score	LDPS	SDPS	IS	F Value	*p* Value
0	97.40 ± 0.74 ^a^	95.50 ± 0.71 ^a^	79.17 ± 5.89 ^b^	18.71	0.023
1	1.56 ± 0.72 ^b^	2.50 ± 0.76 ^b^	10.42 ± 2.95 ^a^	18.48	0.018
2	1.04 ± 0.25 ^b^	2.01 ± 0.12 ^b^	11.52 ± 2.73 ^a^	28.78	0.021
3	0.00 ± 0.00	0.00 ± 0.00	0.00 ± 0.00		
4	0.00 ± 0.00	0.00 ± 0.00	0.00 ± 0.00		

Note: Gait score of Yangzhou geese in the long-distance pasture system (LDPS), short-distance pasture system (SDPS), and indoor system (IS). In the same row, values with different superscripts lowercase letters indicate significant differences (*p* < 0.05).

**Table 8 animals-15-00304-t008:** Frequencies of natural behaviors of geese under different systems.

	LDPS	SDPS	IS	F Value	*p* Value
Standing	7.83 ± 1.17	7.43 ± 2.35	8.17 ± 1.27	0.22	0.744
Lying down	6.34 ± 1.34	7.34 ± 0.89	7.75 ± 0.42	2.36	0.155
Single-foot standing	2.75 ± 0.92 ^a^	2.62 ± 0.53 ^a^	1.00 ± 0.17 ^b^	8.34	0.032
Feeding	8.09 ± 5.92	7.63 ± 3.56	5.75 ± 1.42	0.33	0.542
Drinking	8.75 ± 5.08	7.85 ± 3.35	5.75 ± 2.92	0.94	0.425
Feather pecking	0.75 ± 0.42 ^b^	0.88 ± 0.56 ^b^	3.00 ± 0.83 ^a^	22.10	0.014
Feather combing	14.00 ± 4.67	13.85 ± 3.56	14.59 ± 0.42	0.05	0.839
Wing flapping	2.92 ± 1.09	2.66 ± 1.56	2.17 ± 1.17	0.80	0.460

Note: Frequencies of natural behaviors of Yangzhou geese in the long-distance pasture system (LDPS), short-distance pasture system (SDPS), and indoor system (IS). In the same row, values with different superscripts lowercase letters indicate significant differences (*p* < 0.05).

## Data Availability

All data generated or analyzed during this study are included in this published paper.

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
