# Peer review of "Effects of Free-Range Systems on Muscle Fiber Characteristics and Welfare Indicators in Geese"

_animals, 2025, doi:10.3390/ani15030304_

Round 1
Reviewer 1 Report
Comments and Suggestions for Authors
Comment1
Why choose 28-day-age geese?Whether the weight range of the goslings is too large?
Comment 2
Whether 70 days slaughter time is too short
Comment 3
Are there any other examples of poultry?
Comments on the Quality of English Language114:‘’Yangzhou geese were sacrificed at 70 days of age after a 12-h fasting period. “
Is it inappropriate to use “sacrificed”?
Author Response
Dear reviewer #1:
- Why choose 28-day-age geese? Whether the weight range of the goslings is too large?
Response: We feel sorry for the inconvenience brought to the reviewer. In the production process of Yangzhou geese, the period from day 1 to 28 is the starter period, followed by a rapid growth stage called grower period (from day 28 to 70), and the production process is completed in 70 days. We carried out this experiment from day 28 to 70. During this period, the physiological functions of Yangzhou geese tend to be stable, and they can better adapt to environmental changes, preventing interference with the experimental results. Regarding the weight of goslings, the average weight of male geese at 28 days of age is about 1.5 kg. In order to reduce the variation caused by weight differences, we screened the geese according to the standard of no more than 10-15%.
2.Whether 70 days slaughter time is too short.
Response: We feel sorry for the inconvenience brought to the reviewer. Yangzhou geese are typically marketed at 70 days of age in commercial farming. We selected this age for slaughter, aligning with commercial production, to ensure that the results are both practical and provide a robust scientific basis for optimizing production methods.
- Are there any other examples of poultry?
Response: We gratefully appreciate your valuable comment. Similar studies have been conducted on other poultry species. For instance, in the article "Effects of Free-Range Access on Production Parameters, Meat Quality, Composition, and Taste in Slow-Growing Broiler Chickens", it was found that free-range systems negatively impact slaughter weight, while has a positive effect on meat quality, taste, and composition. However, this research didn’t involve in the aspect of animal welfare. In addition, three articles to elucidate the growth traits, slaughter, carcass and meat quality characteristics and behavioral characteristics of artificial and natural hatched geese in intensive and free-range systems.
- Stadig, L.M.; Rodenburg, T.B.; Reubens, B.; Aerts, J.; Duquenne, B.; Tuyttens, F.A. Effects of free-range access on production parameters and meat quality, composition and taste in slow-growing broiler chickens. Poult Sci. 2016, 95, 2971-2978. doi: 3382/ps/pew226.
- Mehmet, A.B.; Musa, S.; Umut, S.Y. Production traits of artificially and naturally hatched geese in intensive and free-range systems – II: slaughter, carcass and meat quality traits. British Poultry Science. 2016, 58, 166-176. doi:1080/00071668.2016.1261998.
- Mehmet, A.B.; Musa, S.; Umut, S.Y. Production traits of artificially and naturally hatched geese in intensive and free-range systems: I. Growth traits. British Poultry Science. 2017, 58, 132-138. doi:1080/00071668.2016.1261997.
- Mehmet, A.B.; Musa, S.; Umut, S.Y. Behavioral traits of artificially and naturally hatched geese in intensive and free-range production systems. Applied Animal Behaviour Science. 2021, 236, 105273-105273. doi:1016/j.applanim.2021.105273.
- 114: ‘Yangzhou geese were sacrificed at 70 days of age after a 12-h fasting period’. Is it inappropriate to use “sacrificed”?
Response: Thank you for pointing out this problem in this manuscript. We have checked and modified. (Line 118, page 3)
Yangzhou geese were slaughtered at 70 days of age after a 12-h fasting period.
Reviewer 2 Report
Comments and Suggestions for Authors
Dear authors, your study has a clear objective to evaluate the effects of different free-range systems (LDPS, SDPS, and IS) on muscle fiber characteristics, welfare indicators, and behavior of geese. However, the study has some important lacks on some points. IYou can find my comments below:
-The study lacks details regarding indoor housing dimensions (IS), which are critical for replicating the experiment and understanding the conditions influencing the control group. There is no information regarding the dimensions of the indoor housing. It is essential to provide details about the area (in square meters) where the animals are confined.
-There is insufficient validation of the histological and molecular methods used. For instance, the use of Gallus gallus myosin gene sequences for evaluating mRNA levels in geese (Yangzhou geese) is questionable, potentially compromising the accuracy of molecular findings. I must express my doubts regarding the reliability of the mRNA levels and the methods used to detect them. Specifically, using the Beta-actin gene specific to geese (Accs. no: M26111.1) as a housekeeping gene is appropriate; however, analyzing myosin expressions based on genes from Gallus gallus (NM_204587.1; NM_001013396.1; NM_204228.3) is incorrect. Therefore, interpreting the detected mRNA levels as representative of geese is not an appropriate approach. Furthermore, non-specific dimer regions were observed in the primers provided by the authors for the myosin genes.
- While the study states that the muscle fiber composition remains predominantly fast-twitch, the mechanisms underlying this observation are not thoroughly explored. The authors acknowledge that muscle fiber differentiation occurs early in life but fail to link this adequately with their findings.
-The analysis of feather cleanliness and damage appears subjective. Using more objective, quantifiable measures (e.g., image analysis) would increase the reliability of these findings.
-The statistical analyses require more detailed reporting. For instance, effect sizes and confidence intervals for the observed differences would strengthen the validity of the conclusions.
-While the study concludes that free-range systems improve welfare and muscle quality, it does not sufficiently address the economic or logistical challenges associated with implementing such systems, which are crucial for practical recommendations. Please discuss how these findings could influence the adoption of free-range systems in commercial goose production. Are there economic or practical considerations that need addressing?
Author Response
Dear reviewer #2:
1.The study lacks details regarding indoor housing dimensions (IS), which are critical for replicating the experiment and understanding the conditions influencing the control group. There is no information regarding the dimensions of the indoor housing. It is essential to provide details about the area (in square meters) where the animals are confined.
Response: Thank you for pointing out this problem in the manuscript. We have added the dimensions of the indoor housing in Materials and Methods.
The three systems each feature identical indoor sheds, measuring 3.2 × 6.3 meters, and are equipped with feeders and waterers. (Lines 100-101, page 3)
2.There is insufficient validation of the histological and molecular methods used. For instance, the use of Gallus gallus myosin gene sequences for evaluating mRNA levels in geese (Yangzhou geese) is questionable, potentially compromising the accuracy of molecular findings. I must express my doubts regarding the reliability of the mRNA levels and the methods used to detect them. Specifically, using the Beta-actin gene specific to geese (Accs. no: M26111.1) as a housekeeping gene is appropriate; however, analyzing myosin expressions based on genes from Gallus gallus (NM_204587.1; NM_001013396.1; NM_204228.3) is incorrect. Therefore, interpreting the detected mRNA levels as representative of geese is not an appropriate approach. Furthermore, non-specific dimer regions were observed in the primers provided by the authors for the myosin genes. (NM_204228.3)
Response: Thank you for pointing out this problem in manuscript. Particularly, we identified goose myosin sequences with high homology to chicken MYH1A and MYH1B. A total of 8 sequences with over 90% homology were found. Additionally, the goose MYH7B sequence show 94.83% homology with the chicken MYH7B (Table 1). Moreover, we have checked again and confirmed that all melting curves in pre-experiment and experiment results displayed a single peak (Figure 1), indicating that primer dimers were not present. Thus, we believe that using the chicken myosin gene sequence to evaluate mRNA levels in geese (Yangzhou geese) is a valid approach.
Table1:
Homology
Anser Myosin Gene Accession Number |
MYH1A (Gallus gallus) |
MYH1B (Gallus gallus) |
MYH7B (Gallus gallus) |
XM_066980178.1 |
92.94% |
94.26% |
|
XM_066980179.1 |
93.29% |
93.06% |
|
XM_066980180.1 |
92.34% |
92.26% |
|
XM_066980181.1 |
92.37% |
92.07% |
|
XM_066980182.1 |
93.94% |
93.12% |
|
XM_066980183.1 |
92.12% |
91.93% |
|
XM_066980184.1 |
91.91% |
92.18% |
|
XM_066980185.1 |
92.25% |
91.90% |
|
MYH7B (Anser cygnoides) XM_048080687.2 |
|
|
94.83% |
Figure 1:
- While the study states that the muscle fiber composition remains predominantly fast-twitch, the mechanisms underlying this observation are not thoroughly explored. The authors acknowledge that muscle fiber differentiation occurs early in life but fail to link this adequately with their findings.
Response: Thank you for pointing out this problem in manuscript. We have made changes based on your per suggestion. And our modifications are as follows.
Type I, or slow-twitch fibers, are seen in high abundance during the early stages of life. However, the proportion of fast-twitch fibers increases substantially with rapid development. This shift occurs because the oxidative metabolic capacity declines with age, and the body compensates for this metabolic decline by increasing the number of fast-twitch fibers.[28] The housing system and feeding environment in later life stages may not significantly change the type of muscle fibers produced. (Lines 435-441, page 11-12)
- The analysis of feather cleanliness and damage appears subjective. Using more objective, quantifiable measures (e.g., image analysis) would increase the reliability of these findings.
Response: Thank you for your valuable feedback. To display our results objectively, we used ImageJ to analyze the grayscale values of the feathers, and the results aligned with our evaluation of feather cleanliness through scoring. We have added this analysis in the supplement material and provided annotations in the original manuscript. For scoring feather damage, we based the evaluation on the exposed skin area (Table 3). During the experiment, we adhered strictly to this standard to minimize subjectivity and ensure the reliability of the scoring. (Lines 333-335, page 9)
Figure S1: ImageJ was used to analyze the grayscale of Yangzhou goose feather images. A: Back feather gray value, B: Chest feather gray value, C: Wing feather gray value. LDPS, long-distance pasture system; SDPS, short-distance pasture system; IS, indoors system. Statistically significant differences are indicated by different letters.
Table 3. The scoring standard for feather quality in geese.
Score |
Feather Cleanliness |
Feather Damage |
0 |
Completely clean |
Feathers are of good quality and no damage |
1 |
The dirty area is less than 1/4 of the scoring area |
Feathers are damaged but no skin is exposed |
2 |
The dirty area accounts for 1/4 - 1/3 of the scoring area |
The exposed skin area is less than 3cm×3cm |
3 |
The dirty area accounts for 1/3 - 1/2 of the scoring area |
The exposed skin area is more than 3cm×3cm |
4 |
The dirty area is greater than 1/2 of the scoring area |
The skin completely exposed |
- The statistical analyses require more detailed reporting. For instance, effect sizes and confidence intervals for the observed differences would strengthen the validity of the conclusions.
Response: We gratefully appreciate for your valuable comment. We have made changes based on your per suggestion. We have added the F - values to Table 7, Table 8, and Table 9 and made the following modifications:
The Yangzhou goose step counts were subjected to a one-way ANOVA followed by Duncan multiple range test. (Line195-196, page 6)
The significance level is set at 0.05, which corresponds to a 95% confidence interval. The F-value is used to test whether there are significant differences in the fixed effects (free-range systems), while the P-value is used to determine the statistical significance of these differences. Results are presented as mean ± standard deviation, and P < 0.05 considered statistically significant. (Line208-212, page 6)
Among the geese in the IS group, the proportion of geese with a gait score of 0 (able to walk normally) was significantly lower (F=18.71, P < 0.05) than that of geese in the LDPS and SDPS groups. (Line365-367, page10)
- While the study concludes that free-range systems improve welfare and muscle quality, it does not sufficiently address the economic or logistical challenges associated with implementing such systems, which are crucial for practical recommendations. Please discuss how these findings could influence the adoption of free-range systems in commercial goose production. Are there economic or practical considerations that need addressing?
Response: Thank you for pointing out this problem in manuscript. We have rewritten the corresponding section in Discussion.
Global consumer demand drives producers to prioritize free-range systems [18,19]. These systems have a positive impact on the natural behavior of geese, allowing them to move freely and forage in outdoor spaces, which significantly enhances their welfare. Additionally, these systems improve the quality and flavor of goose meat, meeting consumer demands for high-quality poultry products. However, the muscle fiber characteristics and welfare indicators of geese under different free-range systems remain unclear. This study investigated the effects of different systems (LDPS, SDPS, and IS) on the muscle fiber characteristics and welfare indicators (feather quality, walking ability, and behavioral changes) of Yangzhou geese, providing valuable insights for commercial goose pro-duction. (Lines 404-412, page 11)
This research presented in this paper currently provides a reference for improving both meat quality and animal welfare in geese breeding. However, free-range systems may be associated with higher costs, which burden producers and hinder industry growth [37]. It is urgent to explore cost-effective farming systems that maintain both the quality and health of poultry meat while reducing production expenses. Further studies are needed to determine how to achieve a win-win outcome in production. Moving forward, we will focus on the economic cost-effectiveness of free-range systems, including factors such as land costs and equipment maintenance, as well as economic traits like meat production and down yield. (Lines 477-485, page 12)
Reviewer 3 Report
Comments and Suggestions for Authors
This research successfully explores how free-range systems influence muscle fiber characteristics as well as welfare indicators in geese. By examining both aspects, the study significantly contributes to the field of animal husbandry. The incorporation of precise measurements, such as step counts and muscle fiber sizes, strengthens the findings and adds validity to the conclusions drawn. Furthermore, the analysis of behaviors and welfare markers, including feather cleanliness and gait assessments, enhances our comprehension of the effects different rearing environments have on geese, extending beyond mere growth metrics. The conclusions drawn offer valuable insights for practical improvements in poultry production, which can inform subsequent research efforts in this area. Nevertheless, it is important to revise the methods section to provide additional information regarding the techniques used for recording and evaluating these behaviors.
Author Response
Dear reviewer #3:
This research successfully explores how free-range systems influence muscle fiber characteristics as well as welfare indicators in geese. By examining both aspects, the study significantly contributes to the field of animal husbandry. The incorporation of precise measurements, such as step counts and muscle fiber sizes, strengthens the findings and adds validity to the conclusions drawn. Furthermore, the analysis of behaviors and welfare markers, including feather cleanliness and gait assessments, enhances our comprehension of the effects different rearing environments have on geese, extending beyond mere growth metrics. The conclusions drawn offer valuable insights for practical improvements in poultry production, which can inform subsequent research efforts in this area. Nevertheless, it is important to revise the methods section to provide additional information regarding the techniques used for recording and evaluating these behaviors.
Response: Thank you very much for your approval of this research. We have provided additional information in Materials and Methods.
The three systems each feature identical indoor sheds, measuring 3.2 × 6.3 meters, and are equipped with feeders and waterers. (Line100-101, page 3)
To comprehensively evaluate behavior of geese, six cameras were installed in each pen to continuously monitor their activities. These cameras were strategically positioned to capture the entire living area, ensuring that all relevant behaviors were clearly visible and accurately recorded. (Line103-107, page 3)
At 69 days of age, the back and chest feathers of six geese from each group were observed and scored. Feather cleanliness was scored in three parts of the body: the back, wings, and chest and abdomen. The feathers were observed for dirt, dust, feces, etc., and the score was given according to the size of the dirty area. The degree of feather damage was evaluated by evaluating the feathers on the head, neck, chest and abdomen, wings, back, legs, tail and anus, and the exposed skin area of each part was observed and scored. The specific scoring standards are presented in Table 3. To ensure the objectivity of the scoring, each goose is scored by three different observers. (Line153-161, page 4-5)
The goose gait score was modified based on gait stability evaluation standards developed by Jones et al. [16]. At 69 days of age, the gaits of all geese in each group were observed and scored individually. The gait of the geese was observed from 8:30 to 11:00 am and from 2:00 to 4:30 pm using cameras recordings. To ensure objectivity, each system was assessed by three observers during the same time period. The scoring standards are listed in Table 4. Following the gait evaluation, the percentage distribution of each gait grade was calculated for each repetition. (Line164-170, page 5)
The behavioral indicators of geese and the definitions of each behavior were developed and modified based on van Krimpen et al. [17]. These behaviors, including standing, lying down, feeding, drinking, feather combing, feather pecking, wing flapping and single-foot standing, are presented in Table 5. Observations were conducted for 2.5 h each in the morning and afternoon, and the frequency of each behavior was recorded. At 63 days of age, six geese were randomly selected from each group for behavioral observations for one week. Six cameras were used to record the frequency of various behaviors in each group of geese daily from 8:30 to 11:00 am and from 2:00 to 4:30 pm. To avoid the impact of humans or other external factors on geese behavior, only the middle 1.5 h of the video data was selected for observation. Additionally, three observers manually recorded the frequency of each behavior by reviewing the video clips, ensuring consistency and accuracy in data collection. The collected data were then compiled and analyzed to assess the impact of different feeding systems on the behavioral patterns of the geese. (Line174-186, page 5)
Reviewer 4 Report
Comments and Suggestions for Authors
The article “Effects of Free-range Systems on Muscle Fiber Characteristics and Welfare Indicators in Geese” is a study on muscle fiber characteristics and health indicators in geese as a function of the free-range system used. Previously, similar studies on geese have been conducted to examine the effects of different free-range housing systems on growth performance, carcass characteristics and meat quality of geese. I hypothesize that the present manuscript is a continuation and extension of the aforementioned study. Methodologically, the work is organized correctly, all stages and evaluation systems are understandable. The article contains useful information for raising geese, as well as new data on morphological changes in muscle in response to pasture. A few comments arose during the reading of the article, the list of which is presented below.
Introduction.
In general, the introduction is consistent with the title of the article and the conclusions outlined.
1. Line 53: contains a misprint - improves -> improved
Materials and methods:
1. Line 117: It is written that out of 90 growing geese, 6 were randomly selected for further slaughter and collection of histologic samples. Was the sex of the curled individuals taken into account?
2. Lines 116-119: No thigh muscles were sampled for the experiment, was this due to any particular reason?
Results:
As a recommendation, we would like to suggest that the authors add information on quantitative measures of meat productivity, e.g. carcass weight after slaughter. In my understanding, this would help to compile the available data in terms of meat productivity, which could then be recommended to industry.
Discussion:
The discussion points out the relationship between morphological characteristics of muscle fibers and quality indicators of meat, in particular its tenderness. I wonder if the authors plan to continue the work in this direction.
Author Response
Dear reviewer #4:
The article “Effects of Free-range Systems on Muscle Fiber Characteristics and Welfare Indicators in Geese” is a study on muscle fiber characteristics and health indicators in geese as a function of the free-range system used. Previously, similar studies on geese have been conducted to examine the effects of different free-range housing systems on growth performance, carcass characteristics and meat quality of geese. I hypothesize that the present manuscript is a continuation and extension of the aforementioned study. Methodologically, the work is organized correctly, all stages and evaluation systems are understandable. The article contains useful information for raising geese, as well as new data on morphological changes in muscle in Response to pasture. A few comments arose during the reading of the article, the list of which is presented below.
Introduction.
In general, the introduction is consistent with the title of the article and the conclusions outlined.
- Line 53: contains a misprint - improves -> improved
Response: We gratefully appreciate your valuable comment. We have modified ‘improves’ to ‘improved’. (Line 53, page 2)
Materials and methods:
- Line 117: It is written that out of 90 growing geese, 6 were randomly selected for further slaughter and collection of histologic samples. Was the sex of the curled individuals taken into account?
Response: We appreciate the reviewer’s comment about the consideration of sex. In this study, we only selected male geese as research subjects to eliminate errors caused by gender. Male geese usually grow faster than female geese, and the muscle fiber characteristics of male geese may be more prominent. Similarly, male geese and female geese will also differ in behavioral phenotypes and physiological characteristics. Therefore, we only selected male geese to ensure that the differences observed in the experiment are mainly due to the influence of feeding patterns rather than gender.
- Lines 116-119: No thigh muscles were sampled for the experiment, was this due to any particular reason?
Response: Thank you for pointing out this problem in manuscript. The current researches on muscle fibers are mainly focused on thigh muscles, while there are few studies on leg muscles (extensor digitorum longus, soleus, and gastrocnemius). Among consumers, the leg muscles have turn into popular consumer areas. Thus, we mainly explored the effects of different free-range systems on the characteristics of leg muscle fibers.
Results:
- As a recommendation, we would like to suggest that the authors add information on quantitative measures of meat productivity, e.g. carcass weight after slaughter. In my understanding, this would help to compile the available data in terms of meat productivity, which could then be recommended to industry.
Response: We gratefully appreciate your valuable suggestion. The data on meat quality is valuable in providing industry-related insights. In this study, we focused on muscle fiber characteristics and welfare indicators, and the similar research about meat productivity have reported in 'Effects of different free-range housing systems on growth performance, carcass characteristics and meat quality of geese'.
Discussion:
- The discussion points out the relationship between morphological characteristics of muscle fibers and quality indicators of meat, in particular its tenderness. I wonder if the authors plan to continue the work in this direction.
Response: We are very grateful for your professional review of our article. We are also concerned about this issue. The muscle fiber characteristics (thickness of muscle fibers, endomysium, perimysium, and muscle fiber type) have a great impact on meat quality, especially tenderness, but the mechanism behind it is still unknown. In the future, we will further explore the relationship between muscle fiber characteristics and tenderness.
Round 2
Reviewer 2 Report
Comments and Suggestions for Authors
Dear authors,
First, I would like to thank you for your responses and the corrections you made in light of my critiques. While many of your revisions have addressed my concerns, I must express that substituting goose genes with chicken genes in your analyses does not seem appropriate. Although the goose genome information for the relevant genes is available on NCBI, I still believe it is unsuitable to use chicken genes as a substitute. Furthermore, when considering the article as a whole, the analysis based on these gene expressions does not appear to contribute significantly to the overall findings. In my opinion, it might even be more appropriate to omit this non-essential information altogether.
Author Response
Dear reviewer:
Comments: First, I would like to thank you for your responses and the corrections you made in light of my critiques. While many of your revisions have addressed my concerns, I must express that substituting goose genes with chicken genes in your analyses does not seem appropriate. Although the goose genome information for the relevant genes is available on NCBI, I still believe it is unsuitable to use chicken genes as a substitute. Furthermore, when considering the article as a whole, the analysis based on these gene expressions does not appear to contribute significantly to the overall findings. In my opinion, it might even be more appropriate to omit this non-essential information altogether.
Response: We gratefully appreciate your valuable comment. We have omitted the relevant information in the manuscript.

Reviewer 4 Report
Comments and Suggestions for Authors
During the review, the authors corrected the text in line 53 in the introduction section.
The materials and methods sections did not answer the questions posed, and additions to the manuscript text do not contain points of interest (lines 116-119, 117).
Also, carcass weight data were not added in the results section.
Comments that have not been responded to by the authors are presented at the bottom.
Materials and methods:
1. Line 117: It is written that out of 90 growing geese, 6 were randomly selected for further slaughter and collection of histologic samples. Was the sex of the selected individuals taken into account?
2. Lines 116-119: No thigh muscles were sampled for the experiment, was this due to any particular reason?
Results: As a recommendation, we would like to suggest that the authors add information on quantitative measures of meat productivity, e.g. carcass weight after slaughter. In my understanding, this would help to compile the available data in terms of meat productivity, which could then be recommended to industry.
Author Response
Dear reviewer:
We have provided the following responses and additions based on your comments.
Materials and methods:
- Line 117: It is written that out of 90 growing geese, 6 were randomly selected for further slaughter and collection of histologic samples. Was the sex of the selected individuals taken into account?
Response: We appreciate the reviewer’s comment about the consideration of sex. In this study, we only selected male geese as research subjects to eliminate errors caused by gender. (Line 83) Male geese usually grow faster than female geese, and the muscle fiber characteristics of male geese may be more prominent. Similarly, male geese and female geese will also differ in behavioral phenotypes and physiological characteristics. Therefore, we only selected male geese to ensure that the differences observed in the experiment are mainly due to the influence of feeding patterns rather than gender. To make the presentation clearer, we have added relevant information to the Materials and Methods section. (Lines 120, 141, 152, 167)
- Lines 116-119: No thigh muscles were sampled for the experiment, was this due to any particular reason?
Response: Thank you for pointing out this problem in manuscript. The current researches on muscle fibers are mainly focused on thigh muscles, while there are few studies on leg muscles (extensor digitorum longus, soleus, and gastrocnemius). Among consumers, the leg muscles have turn into popular consumer areas. We mainly explored the effects of different free-range systems on the characteristics of leg muscle fibers, so we did not take thigh muscles into consideration.
- Results:As a recommendation, we would like to suggest that the authors add information on quantitative measures of meat productivity, e.g. carcass weight after slaughter. In my understanding, this would help to compile the available data in terms of meat productivity, which could then be recommended to industry.
Response: We gratefully appreciate your valuable suggestion. The data on meat quality is valuable in providing industry-related insights. In this study, we focused on muscle fiber characteristics and welfare indicators, and the similar research about meat productivity have reported in 'Effects of different free-range housing systems on growth performance, carcass characteristics and meat quality of geese'. And the key conclusions from this study have been discussed in the Discussion. (Line381-385, page 10)
‘Our previous research revealed that the free-range system did not significantly influence the dressing percentage, percentage of partially eviscerated weight, and percent-age of formally eviscerated weight in geese. But it notably increased the proportion of leg muscle weight while significantly reducing both the pH value and moisture content [10].’
